# Executive Functions and Language Skills in Preschool Children: The Unique Contribution of Verbal Working Memory and Cognitive Flexibility

**DOI:** 10.3390/brainsci13030470

**Published:** 2023-03-10

**Authors:** Marisa G. Filipe, Andreia S. Veloso, Sónia Frota

**Affiliations:** 1Center of Linguistics, School of Arts and Humanities, University of Lisbon, 1600-214 Lisbon, Portugal; 2Center for Psychology at the University of Porto, Faculty of Psychology and Education Sciences, 4200-135 Porto, Portugal

**Keywords:** executive functions, working memory, cognitive flexibility, language, preschool children

## Abstract

The development of language skills requires a range of linguistic abilities and cognitive processes, such as executive functions (EFs, i.e., a set of skills involved in goal-directed activities which are crucial for regulating thoughts and actions). Despite progress in understanding the link between language and EFs, the need for more research on the extent and directionality of this link is undeniable. This study examined whether specific components of EFs account for a significant amount of variance in language abilities above and beyond gender, age, and nonverbal intelligence. The sample comprised 79 typically developing children attending the last year of preschool (*M_age_* = 64.5 months, *SD* = 3.47). EFs were assessed through tasks that explored three predictor variables: inhibitory control, working memory, and cognitive flexibility. The language outcomes included receptive and expressive language. After controlling for age, gender, and nonverbal intelligence, findings showed that working memory and cognitive flexibility, respectively, explained an additional 16% and 19% of the variance. Inhibition skills did not increase the amount of explained variance in language outcomes. These results highlight the potential added importance of assessing working memory and cognitive flexibility in the prediction of language skills in preschool children.

## 1. Introduction

Developing language skills is critical for meeting the demands and challenges of contemporary societies. Unfortunately, many children do not master language skills as expected, and language impairments during childhood tend to persist throughout development, with lifelong implications for academic, social-emotional, and behavioral functioning [1]. Language is a complex process that requires a range of linguistic abilities and cognitive processes, such as executive functions (EFs) [2]. Furthermore, variables such as gender [3], age [4], socioeconomic status [4,5,6,7], or nonverbal intelligence [8] are also known to affect language development. The present study examined the link between the development of language skills and EFs to determine whether specific components of EFs account for the variance in language abilities in preschool children, while controlling for well-known language predictors, namely, age, gender, and nonverbal intelligence.

EFs are a set of cognitive skills involved in goal-directed activities, which are crucial for regulating thoughts and actions [9]. The several definitions of EFs available in the literature illustrate the diversity of abilities under this umbrella term [10,11]. Research has suggested that EFs include at least three separate but related components [12]: inhibition (i.e., inhibiting prepotent responses), working memory (i.e., storing and manipulating information while performing a task), and cognitive flexibility (i.e., switching between two or more tasks/goals) (Miyake et al., 2000). The development of EFs is critical to the individual’s ability to adapt to changes in the environment efficiently [13,14] and to manage daily tasks [9,15,16]. Therefore, it is not surprising that the development of EFs is related to life achievements, such as school performance [17]. EFs emerge during the first years of life and continue to develop until adulthood [18,19]. Children, in particular, exhibit significant gains in EFs during the preschool years [20]. In fact, several studies have suggested a gradual differentiation of executive components during this period, characterized by the gradual emergence of inhibition, working memory, and cognitive flexibility [21,22,23,24].

It has been reported that EFs gradually differ in structure throughout schooling [25]. There is evidence supporting the unitary nature of EFs in the preschool years [26,27,28,29], with subsequent developmental differentiation between EF components [9,13,30,31]. However, contradictory findings seem to also support the presence of specific components of EFs, even in the preschool years. For instance, Miller et al. [22] suggested a differentiation between two primary EF domains, namely, working memory and inhibition, among preschoolers. In a longitudinal study, Usai et al. [23] found a differentiation between two other EF domains among 5- and 6-year-olds, namely, working memory and shifting. Thus, the development and dissociation of EF components throughout child development is still a matter of debate.

Considering the relevance of EFs to language abilities, previous work has examined the relationship between EFs and language skills in clinical populations. This topic has been of long-standing interest in the field of language sciences and clinical linguistics (e.g., [32,33,34,35,36,37,38,39,40]). For example, research suggested that children with language impairments perform poorly compared to typically developing peers on working memory [33,39], inhibition [38], and cognitive flexibility tasks [41]. Moreover, a robust longitudinal relationship has been found between language and EFs in children at risk for language learning impairments in the transition from preschool to schooling [36]. Botting et al. [42] showed that deaf children performed significantly lower on nonverbal EF tasks than their typically developing peers, even after controlling for processing speed and nonverbal ability [29].

Difficulties in EFs have also been reported in neurodevelopmental disorders often characterized by language difficulties, such as autism spectrum disorders [15] or attention deficit/hyperactivity disorder [43]. For example, Weismer et al. [40] found an association between language and EF skills in children with autism. Filipe et al. [34,35] showed that EFs in children with autism are linked to pragmatic language and prosodic skills. Petersen, Bates, and Staples [44] suggested that language impairments may lead to later inattentive–hyperactive behavior difficulties by affecting self-regulation skills.

The link between language and EFs has also been examined in children with typical development (e.g., [45,46,47,48,49,50]). Kaushanskaya et al. [45] explored the association between performance on nonverbal EF tasks (viz., inhibition, shifting, and updating) and language outcomes in typically developing children, ages 8 through 11. The authors found two critical links: (a) working memory and receptive language and (b) inhibition and syntactic abilities. Still, most studies on the link between language and EFs focused on working memory skills (e.g., [51,52,53]). Performance in working memory tasks, for example, has been reported to trigger auditory and written sentence comprehension in children and adults (e.g., [53]) and sentence production in young adults [48]. Indeed, working memory may be essential to language development, as it temporarily stores information while performing ongoing mental operations on the stored data.

Even though the relationship between EFs and language has been considered in previous studies, research exploring the relationship between language, inhibitory control, and cognitive flexibility is less common. However, these specific domains may also have important roles in language development. For example, inhibitory control may play an essential role in language development, as it may allow children to focus on one of the various possible interpretations of a message [54]. It may also be essential in communicative perspective taking [55,56]. In turn, cognitive flexibility may allow children to use language more flexibly. For example, cognitive flexibility allows adjustable language processing, which depends on selective activation and suppression of linguistic forms and meanings, language cues, task demands, contextual factors, and internal cognitive states [57,58]. Thus, it becomes important to consider the contributions of these EF domains to language development.

### Current Study

Previous research has suggested a connection between EFs and language. However, most studies have been carried out on specific components of EFs and mainly on working memory. There is less research on other executive domains, such as inhibitory control and cognitive flexibility. Moreover, although significant changes in EFs and language characterize the preschool years, few studies have focused on this period. Therefore, despite the general consensus on the link between EFs and language, we know relatively little about the associations between EF abilities, their components, and language skills among preschool-aged children. Importantly, the unique contribution of each EF domain to language skills in the preschool period has not yet been determined.

In the present study, we examined the relationship between working memory, inhibitory control, cognitive flexibility, and language abilities during the last year of preschool. By conducting multiple hierarchical regressions, we aimed to identify the unique contributions of these specific EF components to language outcomes. Given previous findings, we also controlled for age, gender, and nonverbal intelligence, which are known predictors of language development. We selected EF measures commonly used in previous studies to measure these specific EF components. These measures will determine if inhibitory control, working memory, and/or cognitive flexibility account for a significant amount of variance in language outcomes above and beyond gender, age, and nonverbal intelligence. The findings from this study will deepen the current knowledge about the EFs–language link by focusing on language development in preschool children and providing a stringent test for this link through the control of a set of well-known language predictors. Based on previous research and the importance of EFs for language development (e.g., [17,45]), we expect a strong association between EF abilities and language skills. However, given prior contradictory findings on the development of EFs in the preschool years and the link between specific components of EFs and language, no clear predictions about the unique contributions of each EF domain can be made.

## 2. Materials and Methods

### 2.1. Participants

Seventy-nine typically developing European Portuguese children participated in this study (41 girls; mean age = 64.5 months, *SD* = 3.47; age range = 50 to 72 months). The children were enrolled in their last year of preschool (in private schools in a large metropolitan area in the north of Portugal, with high socio-economic backgrounds). Based on the information provided by the nursery teacher, there were no reports of medical conditions or uncorrected visual/hearing problems.

### 2.2. Nonverbal Intelligence Measure

The Raven’s Colored Progressive Matrices [58,59] were used as a nonverbal intelligence estimate. This measure combines three sets, each with 12 items. For each item, children select the missing element that completes a pattern. The final score was the sum of correct answers, and higher scores correspond to better reasoning skills. This test is recognized as a culture-fair test, exhibiting good test–retest reliability and good internal consistency (*r* = 0.80, [60]; Cronbach’s alpha average around 0.85, [61]).

### 2.3. Executive Function Measures

**Inhibitory control.** To assess inhibitory skills, we used the Inhibition subtest of the NEPSY-II, a development neuropsychological assessment [62], which evaluates the inhibition of automatic responses. The subtest includes three components: Naming, Inhibition, and Switching. The combined scaled score of the Inhibition component (i.e., the combination of completion time with errors) was used. This component involves fast opposite naming of shapes (children said ‘square’ when they saw a circle and vice versa) and arrows (children said ‘up’ when they saw an arrow pointing downward and vice versa) [62]. Higher combined scores reflect better performance on this task. This subtest showed good test–retest reliability and excellent internal consistency (*r* = 0.81, [63]; Cronbach’s alpha = 0.92, [62]). 

**Working memory.** Working memory was assessed through the Digit Span subtest from the Wechsler Intelligence Scale for Children-III [64,65]. In this subtest, children are asked to repeat increasingly longer sequences of digits in forward and backward order. The outcome was the number of successfully recalled trials, with higher scores associated with higher short-term/working memory skills. This task exhibited good test–retest reliability and excellent internal consistency (*r* = 0.83, [64]; α = 0.83, [66]). Either forward or backward repetition of digits has been used to measure working memory (e.g., [67,68]). Some studies have indicated that impairments in working memory go undetected when using the digit span backwards and that the digit span forwards reflects the short-term memory component of working memory (e.g., [69]). Therefore, we considered that the overall score would better characterize working memory. For clarification purposes, the term ‘short-term/working memory’, instead of ‘working memory’, will be used throughout the paper.

**Cognitive flexibility.** The Semantic Verbal Fluency subtest of the Coimbra Neuropsychological Assessment Battery (BANC; [70]) was used to assess cognitive flexibility, following Diamond [21]. The subtest evaluates the ability of children to generate words from different semantic categories (viz., animals, names, and food) in 60 s. The final output was the sum of correct words, and higher scores are associated to higher cognitive flexibility abilities. This task exhibited good test–retest reliability and internal consistency (*r* = 0.80, Cronbach’s alpha = 0.76, [71]).

### 2.4. Language Measure

**Language.** To assess expressive and receptive language, we used the Language subscale of the Griffiths Mental Development Scales 2–8 years of age [72]. This subscale evaluates overall language development in tasks such as naming objects and colors, repeating sentences, describing an image, and answering comprehension and similarity/difference questions. This subscale showed excellent internal consistency for this age range (Cronbach’s alpha = 0.92; [72]).

### 2.5. Procedure

During the first term of the academic year (September–December), all children were evaluated in two 45 min individual sessions: one session assessed EFs and the other evaluated language skills. The sessions were held in quiet private rooms and were conducted by trained psychologists who are experts in the field. All the children performed the following tasks: Raven’s Colored Progressive Matrices, Digit Span, Inhibition, and Verbal Fluency in one session; and the Language subscale of the Griffiths Mental Development Scales in the other session. Further tasks were applied, but they are not reported here, as they are not within the scope of the current study. The order of the two sessions was counterbalanced across participants, but the order of the tasks within each session was kept constant.

Ethical approval was obtained by the Ethics Committee of the School of Arts and Humanities of the University of Lisbon (11_CEI2021). The recruitment of participants followed the ethical principles and recommendations of the European Union Agency for Fundamental Rights and the Declaration of Helsinki (developed by the World Medical Association). Written informed consent was obtained from the children’s caregivers.

### 2.6. Data Analysis

Bivariate correlations were run between all variables as a first exploration of the relationship between the different EF components, the language outcomes, age, gender, and nonverbal intelligence. We used multiple hierarchical regression models to identify the unique contributions of the various EF components to language outcomes. These models test and describe the relations between the performance in the tasks of inhibitory control, working memory, and cognitive flexibility (viz., Inhibition subtest–NEPSY-II, Digit Span–WISC-III, and Semantic Verbal Fluency–BANC, respectively) and the outcome in a standardized measure of language (viz., Language subscale–Griffiths Mental Development Scales), above and beyond the covariables gender, age, and nonverbal intelligence (as assessed by the Raven’s Colored Progressive Matrices). The variables entered the model in two steps: Step 1 included the covariates (gender, age, and nonverbal intelligence); and Step 2 contained each EF component separately, together with the covariates. This strategy allowed us to understand the unique and incremental contribution of each EF component above and beyond the selected covariates, statistically assessed by the incremental R^2^ (ΔR^2^) values and associated t-tests.

## 3. Results

The descriptive statistics for all variables are presented in Table 1. The skewness and kurtosis values were within an acceptable range, below |3| and |10|, respectively [73]. Other visual evaluations of normality did not reveal substantial deviations. According to the technical and interpretative manuals, the values for nonverbal intelligence, language, and EFs were within the expected range for the ages under analysis.

Bivariate correlations showed that all variables related to EFs were correlated with one another. Language abilities were correlated with short-term/working memory and cognitive flexibility. No correlations were found between language scores and the selected covariables (gender, age, and nonverbal intelligence) (cf. correlation matrix presented in Table 1). The correlation analysis provided a first indication that the relationship between specific EF components and language outcomes was not similar across EF domains.

We conducted multiple hierarchical regressions with two steps to examine the unique contribution of inhibitory control, short-term/working memory, and cognitive flexibility to language outcomes. Step 1 included the covariables gender (0 = female, 1 = male), age, and nonverbal intelligence as predictors and language skills as the dependent variable. In Step 2, each EF component (viz. inhibitory control, short-term/working memory, and cognitive flexibility) was included separately as the predictor variable (see Table 2). This strategy allowed us to measure the independent contributions of the selected variables that were statistically assessed through incremental *R*^2^ (ΔR^2^) values and associated *t*-tests. The results are presented by EF component. The inspection of the variation inflation factor (VIF) did not show evidence of multicollinearity (VIF < 2) for all analyses.

The covariate model of the regression analysis was not significant, *R*^2^ = 0.04, *F*(3, 74) = 1.06, *p* = 0.371). The addition of inhibitory control in Step 2 did not increase the explained variance in language outcomes, *F*change < 1. However, when short-term/working memory was added in Step 2, the model became significant (*R*^2^= 0.19, *F*(4, 73) = 4.54, *p* = 0.002). Furthermore, there was a substantial increase in the amount of explained variance in language outcomes, ΔR^2^ = 0.16, *F*change(1, 73) = 14.41, *p* < 0.001. Short-term/working memory was a significant and unique predictor, above and beyond the covariables (β = 0.43). The addition of cognitive flexibility in Step 2 (*R*^2^ = 0.23, *F*(4, 73) = 5.34, *p* < 0.001) also contributed to a significant increase in the amount of explained variance in language outcomes, ΔR^2^ = 0.19, *F*change(1, 73) = 17.46, *p* < 0.001. Cognitive flexibility was found to contribute significantly and uniquely to language outcomes, above and beyond the covariables included in the model in Step 1 (β = 0.46). The multiple hierarchical regression analyses thus highlighted two particular EF components as significant contributors to language: short-term/working memory and cognitive flexibility. Inhibitory control, by contrast, showed no relationship with language outcomes.

## 4. Discussion

Despite the growing progress in understanding the link between language and EFs, the current understanding of the relationship between specific EF components and language skills is still limited. The current study aimed to examine the association between specific components of EFs and language abilities in preschool-aged children, as this is a critical period for the development of these cognitive processes. Our approach was designed to overcome the gaps identified in previous research. We focused on the unique contribution of understudied executive functioning components, while controlling for well-known language predictors. Specifically, we tested whether inhibitory control, short-term/working memory, and/or cognitive flexibility would explain a significant amount of variance in preschoolers’ language outcomes beyond gender, age, and nonverbal intelligence. Overall, we found evidence of a relationship between EFs and language skills during preschool age: better performance on short-term/working memory and cognitive flexibility tasks predicted better language outcomes after controlling for gender, age, and nonverbal intelligence. Short-term/working memory and cognitive flexibility, respectively, explained an additional 16% and 19% of the variance. By contrast, inhibition skills did not increase the amount of explained variance in language outcomes.

In line with previous studies (e.g., [51,52,53]), our findings support the link between short-term/working memory and language skills, as an association has been found between a better ability to store and manipulate information while performing a task and better language performance. Furthermore, our results add to previous findings by suggesting a relationship between a higher capacity to change perspective and adapt to environmental changes (i.e., cognitive flexibility) and an increase in language performance during the preschool years. Importantly, although a large body of literature highlights the association between working memory and language skills (e.g., [51,52,53]), our results underline the crucial link between cognitive flexibility and language performance.

Moreover, these findings confirm the differentiation of specific EF components during the preschool years, while also adding key contributions to the developmental path of EF components. Previous studies have provided evidence of two primary domains of EFs during childhood: working memory and inhibitory control [74,75,76]. Our results emphasized the vital contribution of cognitive flexibility. This is in line with reports highlighting that cognitive flexibility also develops during preschool age [77,78,79,80,81,82]. However, contrary to previous findings, our results show a lack of association between inhibitory control abilities and language outcomes. Indeed, previous research with typically developing children under five years of age suggested that inhibitory control abilities might explain language variability (e.g., [83,84]). Šimleša et al. [84] found that verbal working memory and inhibitory control, but not cognitive flexibility, predicted language comprehension in 4- to 5-year-old children. The developmental trajectory of the components of EFs could explain these different findings. Inhibitory control is one of the first components of EFs to develop [25] and could be more important at a young age. Working memory and cognitive flexibility could be more critical for more complex tasks assessed at a later preschool age. Previous research has suggested that the latter components of EFs may be more important for complex problem solving than for simpler tasks in younger children (e.g., [85]). However, drawing robust conclusions based on divergent findings reported across studies is difficult. The differences found might be linked to the implementation of different paradigms for assessing EF skills, among other methodological factors (see [86,87]).

### Limitations and Future Directions

The present work has important limitations. First, since the data were obtained at a single timepoint, causality inferences should be avoided. Further longitudinal research is required to replicate these findings, address the causality question, and ascertain whether different processes influence others over time. Second, although we used traditional performance-based tests to measure EFs, which were selected from theoretically motivated models [21], the results obtained through these measures may reflect optimal performance under controlled conditions and may not be illustrative of the child’s daily functioning [88,89,90,91]. Consequently, future studies should consider including measures with higher levels of ecological validity, such as rating scales. In addition, the complexity associated with the EF construct could be overlooked by our single indicators approach (i.e., one measure to assess one EF component). Thus, it is advisable to cross-validate the results using a multiple-indicator approach in future research. Finally, as we used tasks to assess EFs that present cues and instructions using verbal prompts, the relationship between EFs and language could be partially justified by the overlapping linguistic demands across the two domains [45,84,92]. Future research should also include nonverbal EF tasks to explore the relationships between language, working memory, and cognitive flexibility.

In short, more studies are needed to further advance our understanding of the links between language and EFs. In particular, future studies could include controlled interventions to explore the impact of an EF intervention in children’s language (or vice versa). Research has shown that interventions to improve language skills in children with language difficulties might be effective (e.g., [93,94]). However, there is no evidence of their effects on EFs. Similarly, although different activities have been shown to improve EFs in children [12,95], there is no evidence that their benefits might extend to language. Additionally, studies with clinical populations characterized by difficulties in EFs and language impairments (such as autism spectrum disorders [96]) could provide important evidence on the links between language and EFs. Future studies should thus focus on carrying out longitudinal research and explore the link between EF skills and language abilities in clinical populations.

## 5. Conclusions

The present findings provided insight into the unique contribution of short-term/working memory and cognitive flexibility to language outcomes, above and beyond gender, age, and nonverbal intelligence, in preschool children. By analyzing the relationship between these variables, the current study supports the importance of considering not only short-term/working memory, but also cognitive flexibility, when exploring the relationship between EF components and language outcomes during the preschool period. Thus, our results provide a foundation for further research on EF components associated with the development of language skills. Furthermore, although executive deficits are commonly seen in children with language impairments, our findings suggest that each EF dimension may have a distinct contribution to language development. In the particular case of clinical settings, the present findings support the added importance of assessing specific features of EFs when evaluating children’s language development.

## Figures and Tables

**Table 1 brainsci-13-00470-t001:** Descriptive statistics and correlations between measures.

	Descriptive Statistics		Bivariate Correlations					
Variables	*M*	*SD*	1	2	3	4	5	6
1. Gender (1 = boy, 0 = girl)	0.48	0.50						
2. Age (months)	64.49	3.47	−0.11					
3. Nonverbal intelligence	18.27	4.38	0.03	0.21				
4. Language	83.96	10.46	0.01	0.19	0.11			
5. Inhibitory control	9.42	2.49	0.07	−0.07	0.33 **	0.10		
6. Short-term/working memory	6.68	1.90	0.08	0.14	0.35 **	0.42 **	0.25 *	
7. Cognitive flexibility	25.39	7.94	−0.01	0.33 **	0.14	0.47 **	0.23 *	0.23 *

* *p* < 0.05, ** *p* ≤ 0.001.

**Table 2 brainsci-13-00470-t002:** Parameter estimates for regression models that predict language outcomes.

Predictor	*B*	β	*t*
Step 1 (Covariate model)			
Gender	0.59	0.03	<1
Age	0.53	0.18	1.50
Nonverbal intelligence	0.17	0.07	<1
Step 2 (Executive function models)			
Inhibitory Control	0.45	0.11	<1
Short-term/Working memory	2.42	0.43	3.80 *
Cognitive Flexibility	0.62	0.46	4.18 *

* *p* < 0.001.

## Data Availability

Given the restrictions imposed by the Ethics Committee, the detailed data from this study are available from the corresponding author, MF, upon request.

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
