# Peer review of "Executive Functions and Language Skills in Preschool Children: The Unique Contribution of Verbal Working Memory and Cognitive Flexibility"

_brainsci, 2023, doi:10.3390/brainsci13030470_

Round 1

Reviewer 1 Report

New references are needed

Author Response

Thank you very much for the useful review of our manuscript. In response to your comment, we are submitting a revised version of the manuscript.

A detailed response to the comments is given below. The reviewer comment is in black, and our answer is in blue. In the manuscript, changes are highlighted in dark blue.

  1. “New references are needed”

We added more references in the Introduction and Discussion to provide an in-depth understanding of the research question. Many thanks for the suggestion.

Reviewer 2 Report

Dear author,

Thank you very much for the submission. This research is interesting and provides several factors associated with language skills in children and might have an impact on adults in the long term. Longitudinal research and testing need to be performed and addressed in further studies.

I have two comments according to this research.

1: Please add another paragraph describing the statistical methods and analyses in detail in the Material and Method part.

2: It would be better to add raw data for evaluation in the Supplementary part.

Author Response

Thank you very much for the useful review of our manuscript. In response to your comments, we are submitting a revised version of the manuscript. We have carefully considered the comments and prepared the revised version taking them into account. 

A detailed response to the comments is given below. The reviewer comments are in black, and our answers are in blue. In the manuscript, changes are highlighted in dark blue.

  1. “Please add another paragraph describing the statistical methods and analyses in detail in the Material and Method

We included more detail about the statistical methods and analyses in the revised version, by adding a section on ‘Data Analysis’. Thanks for this suggestion.

  1. “It would be better to add raw data for evaluation in the Supplementary part.”

To address this comment, given the restrictions imposed by the Ethical Committee, the raw data of this study will beavailable from the corresponding author, MF, upon request.

Reviewer 3 Report

The paper is short, too short. 

The English language and written style are acceptable, but not always up to academic standards, and it is clear that the Authors are not native speakers. 

A thorough revision of the language and written style, with the help of a native speaker, would help a lot to improve the clarity of the paper and would make it more professional. 

Typos are noticeable here and there, please, fix them. 

The sample analyzed by the Authors seems large enough to provide indicative results; it is up to the Editors, nonetheless, to decide if the results themselves are significant enough. I find them acceptable, but the size of samples analyzed and the number of subjects in a research are always a delicate issue, prone to interpretation. 

The Introduction is too short. It should be expanded, by stressing again on the research goals of the Authors and by clearly specifying how the Authors plan to achieve them. 

Materials and Methods is a too short section. Explain more comprehensively and better the methodology, please. 

And where is the Literature Review? 'Scattered' between section 1 and section 2? Why not to develop and implement a proper literature review, in a section between section 1 and section 2? That would definitely make the paper more accessible and user-friendly for even non-specialized audiences, and would improve significantly its format and its intrinsic structure. Add more works, also, used and cited by you - and also some more general studies, useful for all readerships -, to complete the literature review and develop it as it should be developed. 

The 'experiment' has to be explained more in-depth, as well as the section with the results. It does not need to be 'too long', but some preliminary comments and a systematization are necessary. 

The discussion should be expanded, with a lot more of analysis and comments, also to show the personal / original points of view of the Authors. The discussion is the 'meat of a paper'. Why is it so short and inconsistent? 

This paper does not have a Conclusion. Well, an academic paper should always have a conclusion. Please, add a conclusion, summarizing, like in a 'mirror' with the Introduction, the research goals of the article and how you have achieved them and underlying the value of the article in itself in the current panorama of studies in its field. That would be an appropriate 'closure' for the article. 

The 'experiment' in itself is acceptable, the findings are even interesting, the approach is, in a way, 'original' (or something like that). 

All in all, the paper is not devoid of interest, but it is hugely incomplete and requires a lot of work and enhancements before being considered for publication. 

Thank you. 

Author Response

Thank you very much for the useful review of our manuscript. In response to your comments, we are submitting a revised version of the manuscript. We have carefully considered the comments and prepared the revised version taking them into account. 

A detailed response to the comments is given below. The reviewer comments are in black, and our answers are in blue. In the manuscript, changes are highlighted in dark blue.

  1. “The paper is short, too short.”

We expanded the paper. The introduction now includes a more in-depth review of the literature and further clarification of the motivation for the current study. The Materials and Methods section now includes a more detailed description of the procedure and a new section on data analysis. The results section was also revised and extended. Finally, the discussion was expanded. The paper now includes a section on limitations and future directions of research and a conclusion.

  1. “The English language and written style are acceptable, but not always up to academic standards, and it is clear that the Authors are not native speakers. A thorough revision of the language and written style, with the help of a native speaker, would help a lot to improve the clarity of the paper and would make it more professional. Typos are noticeable here and there, please, fix them. 

Thank you for your comment. The text was thoroughly revised with the help of a native speaker.

  1. “The Introduction is too short. It should be expanded, by stressing again on the research goals of the Authors and by clearly specifying how the Authors plan to achieve them.” 

As per your and other reviewers' request, we expanded the introduction and further highlighted the research goals and the steps to achieve them.

  1. “Materials and Methods is a too short section. Explain more comprehensively and better the methodology, please.” 

The methods section now includes a more detailed description and a new section on data analysis with details on the statistical methods and analyses performed. Thanks for this suggestion.

  1. “And where is the Literature Review? 'Scattered' between section 1 and section 2? Why not to develop and implement a proper literature review, in a section between section 1 and section 2? That would definitely make the paper more accessible and user-friendly for even non-specialized audiences, and would improve significantly its format and its intrinsic structure. Add more works, also, used and cited by you - and also some more general studies, useful for all readerships -, to complete the literature review and develop it as it should be developed.” 

Thank you for your comment. The introduction now includes a more in-depth review of the literature, including more references comprising specialized and more general studies.

  1. “The 'experiment' has to be explained more in-depth, as well as the section with the results. It does not need to be 'too long', but some preliminary comments and a systematization are necessary.” 

We now explain the study more in-depth, both in the methods section and in the results section, following your suggestions. Preliminary comments and systematization sentences were also included.

  1. “The discussion should be expanded, with a lot more of analysis and comments, also to show the personal / original points of view of the Authors. The discussion is the 'meat of a paper'. Why is it so short and inconsistent? This paper does not have a Conclusion. Well, an academic paper should always have a conclusion. Please, add a conclusion, summarizing, like in a 'mirror' with the Introduction, the research goals of the article and how you have achieved them and underlying the value of the article in itself in the current panorama of studies in its field. That would be an appropriate 'closure' for the article.” 

Thank you for your comments. We expanded the discussion, following your suggestions. We included a section on limitations and future directions of research We also added a separate section at the end of the manuscript with a conclusion.

Round 2

Reviewer 3 Report

The article has been improved. 

  It can be considered for publication.